# An Integrated Assessment of the Environmental and Economic Impact of Offshore Oil Platform Electrification

**Luca Riboldi [1,*]** , **Steve Völler [2]** , **Magnus Korpås [2]** and **Lars O. Nord [1]**

[1] Department of Energy and Process Engineering, Norwegian University of Science and Technology-NTNU, 7491 Trondheim, Norway; lars.nord@ntnu.no

[2] Department of Electric Power Engineering, Norwegian University of Science and Technology-NTNU, 7034 Trondheim, Norway; steve.voller@ntnu.no (S.V.); magnus.korpas@ntnu.no (M.K.)

* Correspondence: luca.riboldi@ntnu.no; Tel.: +47-45071463

**Abstract:** Electrification of offshore oil and gas installations on the Norwegian continental shelf is one of several options to decrease the $CO_2$ emitted from these installations. However, there is an ongoing debate regarding how the increased electricity consumption will influence the $CO_2$ emissions in the power market, both in the short-run and in the long-run. This paper aims to address the issue and investigate the feasibility of the electrification of a large offshore area in the North Sea in comparison to standard concepts to supply energy offshore. A novel integrated model was developed for the purpose that includes and combines a process model of the offshore power generation units and a model of the European power system. The integration of the two models allows to simultaneously simulate the behavior of the offshore energy conversion systems and the effect of electrification on the onshore power system. The outcomes of the analysis show that the environmental performance of electrification is strongly affected by the selected approach to quantify the $CO_2$ emissions associated with power from shore. Taking standard methods to supply offshore energy as basis for comparison, the marginal effect of electrification would result in increased $CO_2$ emissions (+40%), while the average effect would entail large reductions in $CO_2$ emissions (−48% to −90%), the extent of which depends on the geographical scope selected. An analysis on the economics of electrification indicates that its economic viability would be challenging and would not be favoured by a strong European commitment towards environmental policies since the expected increase of power price will outbalance the gains for the reduced emission costs.

**Keywords:** oil & gas; energy supply; power from shore; power system; $CO_2$ emissions

---

## 1. Introduction

An increasing awareness of the global warming issue drives governments to seek viable options to decrease their national greenhouse gas emissions. Even though we are witnessing a rapid surge in the contribution of renewable energy sources, it is generally acknowledged that fossil fuels will still play a role in the upcoming decades [1]. It is thus sensible to look into ways to reduce the environmental impact associated with fossil fuel extraction and utilization in order to guarantee a sustainable transition to cleaner alternatives. Norway is in a peculiar situation. The petroleum industry is Norway's largest industry and Norway was in 2017 the 8th largest producer of oil and the 3rd largest producer of gas in the world [2]. The vast majority of the oil and gas produced is exported, meaning the emissions related to the exploitation of those energy sources do not directly weigh on the national account. Nevertheless, the oil and gas extraction requires energy-intensive processes, leading to emissions to the atmosphere. The sector was responsible for 14.8 Mt $CO_2$ equivalents in 2016 [3], representing about 28% of the national emissions. It is clear that any plan for the reduction of the

total emissions in Norway cannot avoid tackling the petroleum sector and, in particular, the offshore activities. Several energy efficiency measures have been proposed to reduce the carbon footprint of offshore installations [4]. The largest share of emissions derives from the local utilization of a fraction of the produced gas to fuel gas turbines. The utilization of onsite gas turbines to supply power and heat is well established. However, the gas turbines are often operated with poor efficiency and many studies indicated their exhaust gases as a main source of exergy loss (see for example [5]). The most obvious countermeasure is to exploit the waste heat available through some waste heat-to-power configurations. There is a comprehensive literature on the topic, including an analysis of the challenges and limitations of offshore bottoming cycles [6], analyses on the best working fluid [7], optimal design of combined cycles (an organic Rankine cycle for power generation in [8] and for combined heat and power in [9], a steam Rankine cycle for power generation in [10] and for combined heat and power in [11]) also considering off-design operating conditions [12], site-scale integration considerations [13], and evaluation of dynamic operation and of control strategies for fast load change [14]. Even though offshore combined cycles are proved to be effective for achieving $CO_2$ emissions reductions, their utilization has been very limited. On the Norwegian continental shelf only three such projects were developed [15], as concerns regarding the additional weight, the complexity of the power cycle, and the cost prevailed. A number of other options were more recently proposed and investigated, involving the power plant [16] as well as the processing plant [17]. A way to decarbonize the offshore operation without deeply modifying the power generation system could be to introduce a carbon capture process [18]. CCS could also allow a concept involving offshore production of clean power from gas and export of the surplus to the mainland to help decarbonise mainland electricity [19]. Alternatively, the integration of wind power to either simple cycle gas turbines [20] or combined cycles [21,22] could contribute to cut $CO_2$ emissions, though not to eliminate them as a conventional power generator is still needed to provide base load and heat to the processes. The integration of renewable sources could as well be a far-sighted option as it could allow reconverting the oil & gas installations into renewable energy production sites [23]. Once the power demand from the petroleum extraction ceases, a number of innovative concepts could be envisaged to utilise the wind power, for instance the synthesis of hydrogen, methane or ammonia [24]. Another possibility that has obtained strong political support in Norway involves the electrification of the offshore facilities. The concept relies on the assumption that power could be provided from onshore renewable sources or generated more efficiently onshore by large thermal power plants. Technical and economic analyses of offshore electrification have been presented [25]. The environmental performance of electrification tends to be analysed with a simplistic approach, for instance associating average values of $CO_2$ emissions to the power sent offshore [17]. However, the power system is a very integrated system that is predicted to change deeply in the upcoming years. The effect of a local increase of the power demand over the years has to be evaluated with appropriate models. Moreover, several methods exists to allocate $CO_2$ emissions to the power supplied to the various consumers. The selection of one method over the others can result in very different outcomes [16]. Similarly, the economic performance of electrification is affected by the medium-to-long-term changes in the energy and environmental policies that have an effect on the cost of fuels, electricity and emissions. The economic analyses available in the literature fall short of accounting for these effects. This work aims to take into consideration all the outlined complexities associated with the electrification of an offshore area and so to provide a comprehensive assessment on the effectiveness of the concept. The methodology developed for the work involves the integration of a process model describing the operation of the offshore power generation with a detailed model of the European power system. The integrated model, coupling the power system model and the process model, allows to evaluate the impact of electrification at different levels, i.e., locally on the offshore installation and globally on the power system. Further, an effort is made to provide a reasonable evaluation on the evolution of the power system during the estimated lifetime of the offshore project so to account for the expected long-term effects. To the authors' knowledge, such a type of approach is otherwise missing in the literature and is, thus, expected to add to the body

of knowledge on the topic. The ultimate objective is to evaluate the feasibility of electrification with respect to standard concepts, namely the utilization of gas turbines. The environmental performance is the main metric used to compare the different concepts, though indications on the economic viability are also provided. A case study was selected at the basis of the analysis, namely the Utsira High area. The Norwegian government demanded the operators involved in the offshore area to develop the fields with power from shore. In accordance with this decision, a two-phase development project was defined. Power will start to be sent offshore from 2019 and a full electrification of the area is expected to be achieved from 2022. Given that the decision on the electrification has been already taken, this work aims to give a contribution in hindsight, answering the question: "Was electrification the most effective concept to supply energy to the Utsira High area?"

The paper is structured as follows: the analysis framework is presented, through a description of the case study (Section 2) and of the concepts to be investigated and compared (Section 3); the methods for the analysis are then outlined (Section 4) by presenting the models developed, the approaches selected for the evaluation of the $CO_2$ emission factor, the future scenarios for the energy system and the approach to the economic analysis. The results from the integrated analysis are reported and analysed (Section 5) also by means of a sensitivity analysis (Section 6). Some concluding remarks are finally provided (Section 7).

## 2. The Offshore Area

The Utsira High area was selected as our case study. This large offshore area is constituted by four production fields in the North Sea, about 180 km west of the city of Stavanger, namely Edvard Grieg, Ivar Aasen, Gina Krog and Johan Sverdrup. The first three fields are already producing and have a 20-year expected lifetime. The last is one of the largest discoveries made in the Norwegian Continental Shelf and is predicted to start production in 2019. The production is expected to last for nearly 40 years. The overall area's lifetime is thus considered to be between 2019 and 2058. Large amounts of power and heat are necessary to operate the offshore installations. In first approximation, power is primarily needed for pumps and compressors while the main consumer of heat is the oil stabilization process. Figures 1 and 2 show a prediction of Utsira High area power and heat requirement throughout the fields' lifetime. The aggregated power and heat demands profile were developed through the information retrieved from the field development reports of Edvard Grieg [26], Ivar Aasen [27], Gina Krog [28] and Johan Sverdrup [29]. The annually averaged data are used for the lifetime simulations.

The Utsira High area has been assessed to be well suited for electrification due to the distance from shore (approximately 200 km), large power requirements for the four fields and water depth (100–120 m). The electrification project, divided into two phases, will be implemented, starting from 2019. Johan Sverdrup will get power from shore via two DC cables while Edvard Grieg, Ivar Aasen and Gina Krog will be tied-in with AC cables and electrified from Johan Sverdrup.

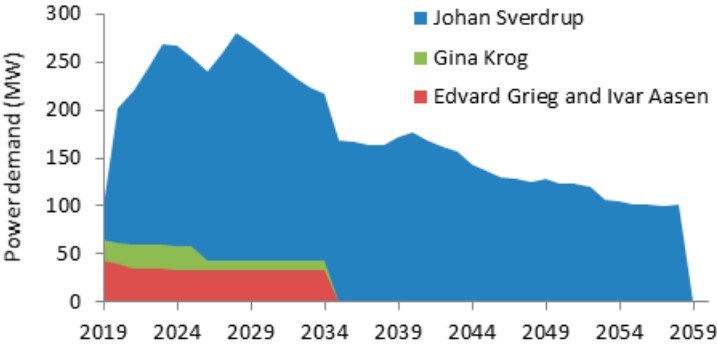

**Figure 1.** Power demand profile of the Utsira High area.

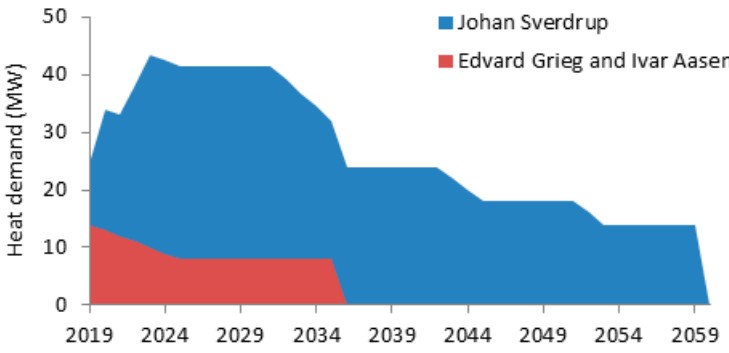

**Figure 2.** Heat demand profile of the Utsira High area.

## 3. Concepts for Offshore Heat and Power Supply

The alternative concepts to supply heat and power to the Utsira High area are described hereafter:

*Concept 0—Gas Turbine Cycles*. This is the common layout for offshore installations. It consists of simple gas turbine (GT) cycles. The power generated by the GTs covers the power demand, while the process heat is supplied by means of waste heat recovery units (WHRUs) exploiting the thermal energy available in the GT exhaust gas. Each installation is equipped with an independent power generation system (GTs + WHRUs) making it energy autonomous (with the exception of Ivar Aasen which was developed to be powered from Edvard Grieg). In order to be able to supply heat and power in each instance of the plants' lifetime, Edvard Grieg is equipped with two GE LM2500+G4, Gina Krog with one GE LM2500+G4 and Johan Sverdrup with six GE LM6000 PF, for a total of nine GTs (3 x GE LM2500+G4 and 6 x GE LM6000 PF). The load allocation strategy between the GTs of each platform considers to split equally the total load between the operating GT, as it is common practice for this kind of applications. Figure 3 illustrates a scheme of the concept.

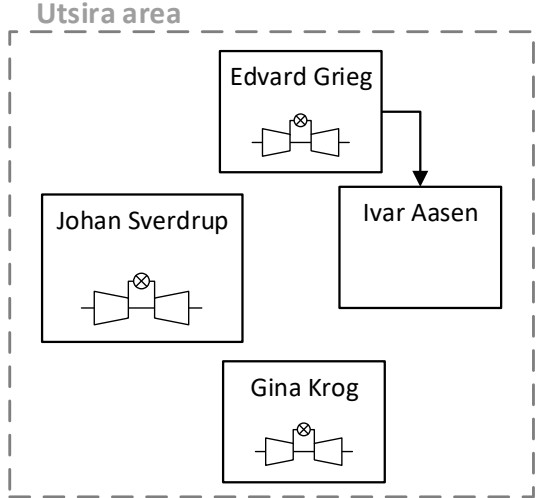

**Figure 3.** Scheme of *Concept 0*—Gas turbine cycles.

*Concept 1—Full Electrification*. This concept involves the electrification of the Utsira High area, with power taken from the onshore grid and the heat locally generated through both gas-fired burners and electric heaters (as indicated in the actual project development plan of the offshore area). The electrification concept is defined in accordance with the actual electrification project to be developed in the area. The various offshore platforms are considered to be tied-in with AC cables. The utilization of power from shore (PFS) necessarily decreases the amount of gas burned locally. Therefore, a larger amount of gas has to be compressed and exported. Such amount is evaluated by taking *Concept 0* as reference case. An additional power consumption to compress this gas is considered. Figure 4 illustrates a scheme of the concept.

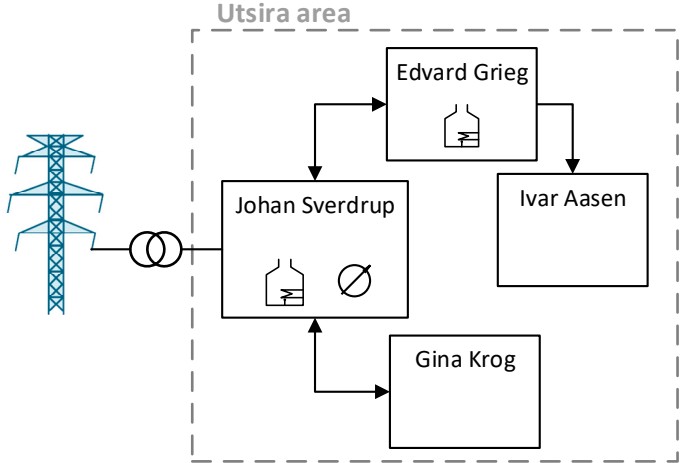

**Figure 4.** Scheme of *Concept 1*—Full Electrification.

*Concept 2*—**Partial Electrification and Gas Turbine Cycles**. This concept is a hybrid of the two previous ones. The heat and a fraction of the power are produced locally by means of GTs and WHRUs, while the remaining power demand is provided by PFS. The various offshore platforms are considered to be tied-in with AC cables. The main role of the GTs is to meet the heat demand, thus the GTs are located on the platforms requiring process heat (i.e., Edvard Grieg and Johan Sverdrup) and their number is the minimum to enable the heat supply in each operating conditions. Three GTs were deemed as sufficient (1 x GE LM2500+G4 and 2 x GE LM6000 PF). The loads at which to operate the GTs is the result of a constrained optimization process: the share between the power produced offshore and the power taken from the onshore grid is optimized so to minimize the $CO_2$ emissions. Figure 5 presents a scheme of the concept.

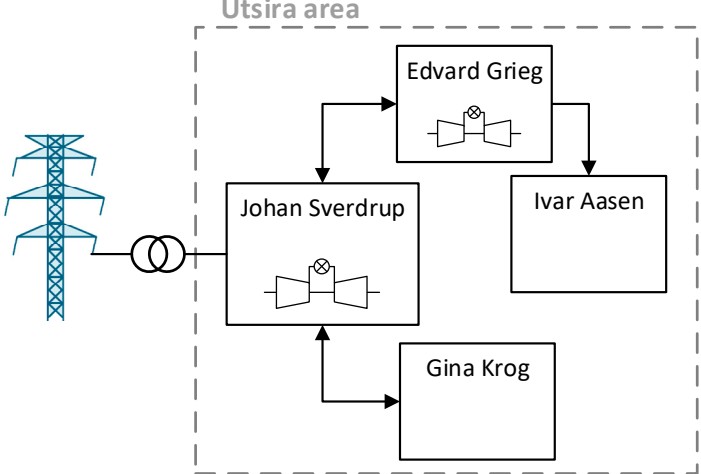

**Figure 5.** Scheme of *Concept 2*—Partial electrification and Gas Turbine Cycles.

## 4. Methods

The process model and the power system model are described in this section. An interchange of information was established to develop the integrated assessment. The power demands from the offshore processes were first fed as inputs to the process and power system model. In the cases involving electrification (full or partial), simulations of the power system are performed. Certain outputs of the power system model ($CO_2$-factor and power spot price) are conveyed to the process model to evaluate the amount of power and heat to be produced onsite. The set of results of the two models are then collected and integrated to simulate the Utsira offshore area performance at different conditions for the concept considered. Figure 6 gives a simplified representation of the develop method.

In this section, the process and power system models are presented. Indications on the different approaches to calculate $CO_2$ factors are as well provided, followed by a description of the principles used to develop long-term scenarios and economic analyses.

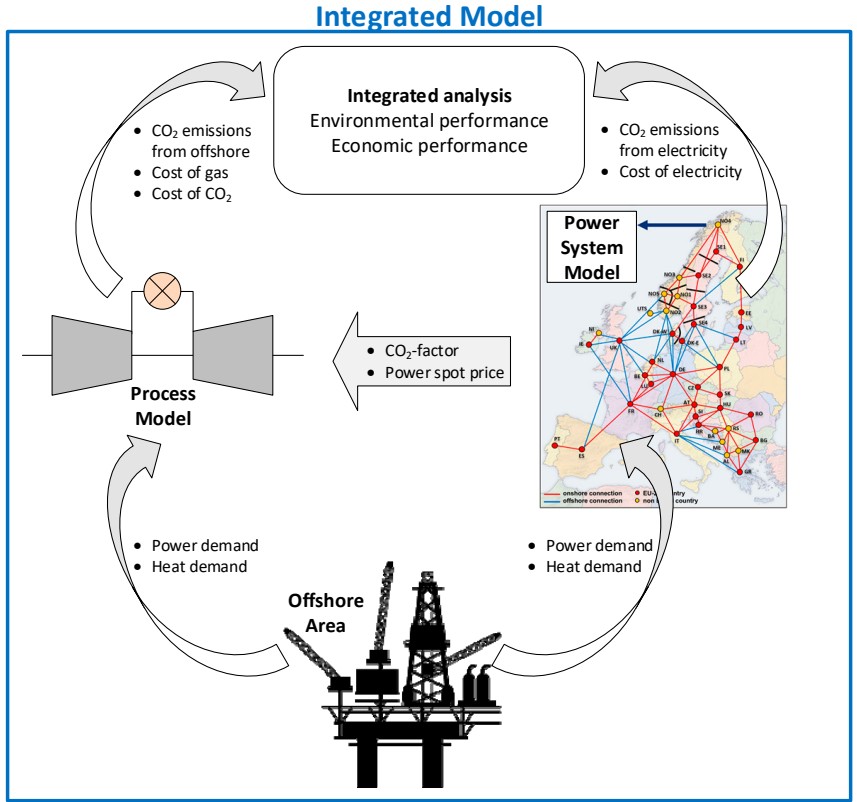

**Figure 6.** Representation of the integrated method developed for the analysis.

*4.1. Process Modelling*

The process models developed were:

**Offshore power plant.** The gas turbines (GT) were modelled with a data-defined model based on the curves provided by the manufacturer. Data for the entire operating range of the engines (10–100%) were retrieved for the two GTs modelled in this paper, a GE LM2500+G4 and a GE LM6000 PF. Those are aero-derivative gas turbines, typically used for offshore applications. The models were implemented in MATLAB (Matlab R2015a, The MathWorks Inc., Natick, MA, USA, 2016) [30] and the performance validated against the same engines in the Thermoflow library of gas turbine engines [31]. Thermoflow indicates that the maximum model errors for the two engines are lower than 0.5% for the exhaust mass flow rate, the power output and the heat rate and lower than 2.8 °C for the exhaust temperature (test range for ambient temperature: −18 to 49 °C). The successful validation suggests that similar uncertainty levels apply also to the models used for this study. A further validation process was carried out for the GE LM2500+G4. The reliability of the model was evaluated by verifying that the simulated performance was in good compliance with the actual performance of the same type of engine installed in Edvard Grieg.

The model of the waste heat recovery units (WHRU) were developed in THERMOFLEX (Thermoflow Inc., version 26.0, Fayville, MA, USA, 2016 [31]), a fully-flexible program for design and off-design simulation of thermal systems. The exhaust gas from the GTs flows counter-currently in the WHRU, designed as a vertical finned tube heat exchanger with staggered tubes and solid fins. Inside the tubes, pressurized (22 bar) water is warmed up to the final temperature (170 °C) required by the heating system. The return water is at 120 °C. The geometry (fin sizes and spacing, and tubes

sizes and spacing) was defined in order to reproduce the WHRU in Edvard Grieg and the simulated performance successfully compared to that.

THERMOFLEX was further used to develop models of gas-fired burners (85% efficiency) and air blowers (85% efficiency). When a surplus of gas was made available on the offshore installations, a gas compression process was modelled with an isentropic efficiency of 85%. The system efficiency for electrical heaters was set at 95%, taking into account transmission and heat losses in the heater. The natural gas available on the offshore installation was the fuel used by the GT and the gas-fired burners. A composition representative of a typical extracted gas in the North Sea was considered. The site conditions were selected to average a year in the specific North Sea geographical location. Table 1 sums up the main assumptions used in the modeling.

**Table 1.** Site conditions used and main modelling assumptions.

| Site | | | Gas Turbines | |
|---|---|---|---|---|
| Ambient T (°C) | 9.4 | | GT fuel | Production gas |
| Ambient P (bar) | 1.013 | | LHV (MJ/kg) | 47.4 |
| Frequency (Hz) | 60 | | GT inlet $\Delta$p (mbar) | 10 |
| Cooling water system | Direct sea water cooling | | GT exhaust $\Delta$p (mbar) | 10 |
| Cooling water T (°C) | 10 | | **Waste Heat Recovery Unit** | |
| **Natural Gas (%vol.)** | | | Tube material | T11 |
| $CH_4$ | 72.9 | | Fin material | T409 |
| $C_2H_6$ | 13.6 | | Fin type | Solid |
| $C_3H_8$ | 8.3 | | Tube layout | Staggered |
| $N_2$ | 1.6 | | **Water Loop** | |
| $CO_2$ | 0.2 | | Inlet water T (°C) | 120 |
| n-$C_4H_{10}$ | 1.8 | | Outlet water T (°C) | 170 |
| i-$C_4H_{10}$ | 0.9 | | **Electrification** | |
| n-$C_5H_{12}$ | 0.3 | | Transmission losses | 11% |
| i-$C_5H_{12}$ | 0.3 | | Transformer efficiency | 99% |
| $C_6H_{14}$ + | 0.1 | | **Gas-Fired Heater** | |
| - | - | | Efficiency | 85% |
| - | - | | **Air Blower** | |
| - | - | | Isentropic efficiency | 85% |

**Power from shore**. When power was sent to the offshore installations, the amount of power to be taken from the onshore grid was augmented due to losses. A 99% transformer efficiency and an 11% transmission loss term were taken into account. The latter is affected by the distance between the power generation site and the offshore site, by the power duty of the platform and by the type and geometry of the power transmission cable.

### 4.2. Power System Modelling

A European power system model was developed to quantify the effect of electrification for future scenarios of the Norwegian and European power system. As a starting point, the EMPS (EFI's Multi-area Power-market Simulator) model was used, which was further developed and customized to meet the modelling requirements of this project.

EMPS was developed by SINTEF Energy Research [32] in the 1970s and has been further developed continuously since then. It is a well proven and established model, which is capable of simulating the whole power system and is used by around 200 users for strategic analyses and price forecasting in the power market. These users include, e.g., transmission system operators, power producers, regulators, consulting companies and academics and research institutions in Norway and abroad. The basic description of the model defines the contents of an area, e.g., watercourses, thermal power plants, PV, wind, different demand curves. The aim of the model is to minimize the expected cost in the whole system regarding all constraints, or, in other words, to maximize the socio-economic benefit. In principle, this solution will coincide with the outcome of a well-functioning (=ideal)

electricity market. These areas are connected via transmission lines to build the model of the energy system. The main applications of the model are long-term operational scheduling of hydropower and forecasting of electricity prices and hydro reservoir operation. The tool's strength lies within the detailed modeling of the watercourses for the Nordic system and the stochastic optimization formulation of the scheduling problem. It takes into account annual variations for climatic distinctions (rainfall, wind, solar radiation) for geographical areas, which, in the case of this work, span over 75 years. This gives a good spread for e.g., hydropower inflow, which influences particularly the hydropower production.

The power system model used for this study builds on the EMPS model. The same framework and approaches used in the well proven EMPS model applies to the extended power system model, allowing to build confidence in its reliability. To be able to extend the scope of the model, the European power system design (core structure)—in terms of power plant fleets, transmission lines, area description, etc.—was embedded, based on the information available from SUSPLAN-project [33]. The "EU Reference Scenario 2016" from the European Commission [34] was used as basis for the model data. The source provides a detailed description of the European energy system from the year 2000 up to 2050, including all EU-28 member states. In a 5-year time-step, information such as installed capacity and produced electricity by different sources are given in a country-wise format. In addition, data about energy system efficiencies, $CO_2$-emissions, CCS-usage or CHP-share is presented. All this information was included in the presented EMPS-scenarios, such as the modelled years depicts the given energy system by the EU reference scenario. Some information was not available in the data given by the European Commission, such as transmission line capacities. In addition, the data were only given for the EU-28 member states, but countries like Norway or Switzerland are not part of it. To include such countries and fill the gaps, data from ENTSO-E's "Ten Year Network Development Plan 2016" are used [35]. For data outside the years given by the sources (2055 and 2060 for the EU data; earlier than 2020 and later than 2030 for the ENTSO-E data), an approach based on extrapolation was used relying on the available information from [33,35]. The price trajectories of fossil fuels and $CO_2$ for the different scenarios considered were taken from IEA [1].

The final model (schematically represented in Figure 7) contains 43 areas, up to 97 transmission lines (depending on the simulation year), 787 thermal power plants of 17 different types, 42 hydropower plants and 103 solar and wind power plants. The power plants were divided into different efficiencies, representing various ages of the plants (old, medium old, modern). With increasing simulation years, the average power plant efficiencies are adjusted according to the average power plants efficiencies given by [34]. Furthermore, several gas power plant types are included in the model (e.g., OCGT, CCGT), with the old ones gradually phasing out over the years. CCS is included in the countries and years as given in [34].

The result of the simulation is an optimal (socio-economic) production mix to cover the electricity demand of all the areas within the given constraints (e.g., transmission and production limits), for each of the climate years. The average result of all climate years is taken further for analysis in this work. Selected output information (production mix, $CO_2$-emissions, power spot prices) of the different scenarios and operational years (2019 to 2058) were inputs to the integrated model.

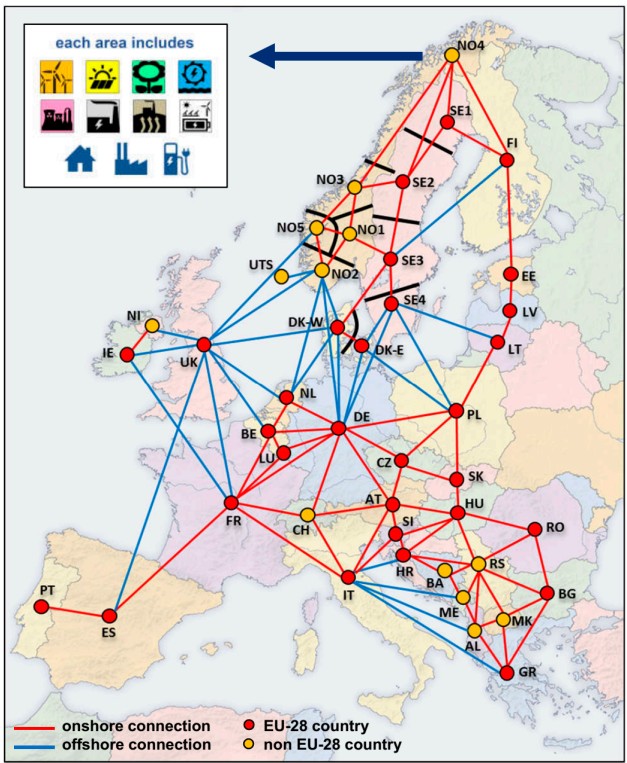

**Figure 7.** Schematic design of the full model in EMPS with the countries/areas and connections.

### 4.3. CO$_2$-Factor

The selection of methods to estimate the CO$_2$-factor associated with PFS is of paramount importance to capture the environmental effect of electrification. The CO$_2$-factor is in general a function of the generator technologies and their respective energy efficiencies and fuel types. Only CO$_2$ emissions related to the use of fuels were considered, i.e., CO$_2$ emissions related to extraction of fuels, constructing of turbines and so on were excluded. For the calculation methods, any emission caps in the EU ETS system were not explicitly taken into account.

The estimation of a CO$_2$-factor addresses the following question "What are the CO$_2$ emissions associated with the electricity supplied to the offshore platforms from shore?" While the several approaches available return very different results [16], the choice of a proper method is often case specific. Two methods were considered in this study, namely the marginal and average CO$_2$-factors. The marginal CO$_2$-factor represents the additional short-term power generation that must be provided by the system, when adding the electrification demand. When the marginal effect is considered, the impact of the additional demand has to be evaluated in terms of additional CO$_2$ emissions with respect to a base case without the new demand (with all other parameters held equal). Hence, it involves the simulation of the power system at two different states. The CO$_2$ emissions factor is calculated as the ratio between the marginal increase of CO$_2$ emissions and the additional demand:

$$\chi_{CO2,\text{marginal}} = \frac{\Delta CO_2 \text{ emissions}}{\Delta \text{power demand}} = \frac{(CO_2 \text{ emissions})_{el} - (CO_2 \text{ emissions})_{ref}}{(\text{power demand})_{el} - (\text{power demand})_{ref}} \tag{1}$$

The average CO$_2$-factor, on the other hand, captures the CO$_2$ emissions associated with the electricity mix at a specified geographical level. The approach assumes that the CO$_2$ emissions should be evenly distributed among all the existing and new power consumers included in the system. The CO$_2$-factor is calculated as the average emission factor of the electricity mix within the system boundaries considered. Four different geographical scopes were used in this study: Norway; Nordic

countries (Norway, Sweden and Finland); Norway and neighbours (Norway, Sweden, Denmark, Germany, The Netherlands and United Kingdom); Europe:

$$\chi_{CO2,\text{average}} = \frac{total\ CO_2\ emissions}{total\ power\ demand} \tag{2}$$

In the very short-run, an increase in power demand due to offshore electrification in the Norwegian Continental Shelf will actually lead to more power production from regulated hydro power, since hydro power dominates the Norwegian generation mix. This again leads to less stored water in the reservoirs, but after a while the reservoir balance needs to be restored by increasing the power production from thermal power generators in neighbouring countries, so in the short run, an increase in the consumption in Norway leads to an increase in thermal power production. This is the marginal effect. Such approach has a potential problem in the identification of what is the "marginal demand". Similarly to offshore electrification, other additional power demands (e.g., a new housing area, new electric cars, even a new and bigger panel oven in the living room) could be defined as "marginal demands". The average method, assigning the same emission factor to each power demand, is a way to overcome this issue. However, it has other disadvantages. A main one is that the results are strongly influenced by the geographical scope in which the average factor is calculated. Defining a relevant boundary in a highly interconnected and dynamic system such as the European energy system is a challenging task. The two methods apparently are very different, and it can be discussed which gives the most correct answer. In this study the outputs based on both approaches are reported. This allows to quantify the spread of results, and to gain a better understanding of the consequences of the assumptions made.

*4.4. Scenarios for the Long-Term Evaluation of the Energy System*

The analysis aims to encompass the expected lifetime of the offshore area, approximately 40 years. If reliable results are to be obtained for such long time-span, it is necessary to estimate the variations of the main inputs to the models. This is achieved by developing long-term scenarios of the energy system. Such process is surely characterized by a significant uncertainty. Energy systems and offshore activities are extremely complex systems, which can be affected by a number of factors. To address the issue, the most reliable datasets available in the literature were used as inputs to the analysis. Moreover, in addition to the scenario selected as basis for the analysis, two other scenarios are introduced. Those are developed as extreme cases and the obtained outcomes help showing the spread of the results that can be expected.

The main input data for the power system evolution (i.e., installed capacities, demand, generation) is taken from the "EU Reference Scenario 2016" [36]. This is a study of a possible future energy scenario within the EU-28 member states, based on historical data starting at the year 2000 and calculated up to the year 2050. The estimation of fuel and $CO_2$ prices throughout the years is based on the World Energy Outlook (WEO) 2016 by IEA [1]. The report includes three different scenarios that describe three possible visions of future prices. The so-called New Policies (NP) scenario is selected as the basis for the analysis. The NP scenario depicts the expected evolution of the prices according to policies and strategies the governments plan to pursue. When a sensitivity analysis is carried out on the influence of fuel and $CO_2$ prices, two additional scenarios are considered, namely the Current Policies (CP) scenario and the 450 (ppm) scenario. Those scenarios reflect, respectively, a world not implementing any new policies (CP scenario) or implementing a series of policies targeting an average global temperature increase within 2 degrees Celsius above pre-industrial levels in 2100 (450 scenario). The scenario price data are available for the period included between 2020 and 2040 with a 10-years interval. The values for the other years are obtained by linear interpolation and extrapolation. The price evolution for the offshore $CO_2$ emissions has to take into account not only the cost of an emission allowance in the EU ETS system but also the $CO_2$ tax that the Norwegian oil & gas sector is subjected to. The policy of the Norwegian government has been to adjust the taxation level in order to retain approximately constant

the overall cost associated with a unit of $CO_2$ emitted. Accordingly, a similar mechanism is applied in the estimation of the total offshore $CO_2$ prices. The price is maintained stable at the current level throughout the years until the WEO 2016 value alone overpasses that threshold. At that point in time, the value of the WEO 2016 scenario is considered. The outcome is a $CO_2$ price that is constant for some initial years, followed by an increase, the extent of which depends upon the WEO scenario considered.

### 4.5. Economic Analysis

The economic analysis is based on the principles of the net present value (*NPV*) method. Given a discount rate (*r*), the cash flows are discounted according to the following formula:

$$DCF_i = \frac{CF_i}{(1+r)^i} \tag{3}$$

where *DCF* is the discounted cash flow, *CF* is the cash flow and *i* is the year when the cash flow occurs. The capital investment is assumed to be made in 2017 in all the cases studied. The discount rate is set to 7%. The annual expenditures considered are due to the onsite gas consumption ($CF_{gas}$), to the cost of $CO_2$ emissions ($CF_{CO2}$) and to the purchase of electricity from the onshore grid ($CF_{PFS}$):

$$CF_i = CF_i^{gas} + CF_i^{CO2} + CF_i^{PFS} \tag{4}$$

The three *CF*s are calculated with the following formulas:

$$CF_i^{gas} = \dot{m}_{gas} LHV_{gas} c_{gas} h_{eq} \tag{5}$$

$$CF_i^{CO2} = \dot{m}_{CO_2} c_{CO_2} h_{eq} \tag{6}$$

$$CF_i^{PFS} = PFS \cdot c_{PFS} \cdot h_{eq} \tag{7}$$

where $\dot{m}_{gas}$ is the mass flow rate of gas used as fuel, $LHV_{gas}$ is the lower heating value of the gas, $c_{gas}$ is the gas price, $\dot{m}_{CO2}$ is the mass flow rate of the emitted $CO_2$, $c_{CO2}$ is the $CO_2$ price, *PFS* is the power taken from shore, $c_{PFS}$ is the power price and $h_{eq}$ are the equivalent hours per year.

The estimation of the capital investment (*CAPEX*) associated to each concept would theoretically allow to calculate the *NPV*:

$$NPV = CAPEX + \sum_{i=1}^{N} DCF_i \tag{8}$$

However, the estimation of the *CAPEX* demonstrated to be a complex task, especially for concepts involving electrification. A large amount of information would have been needed, which was not all available. Other studies discussed how significant variations can be expected due to factors such as financing conditions, market conditions and geographical location [25]. Rather than having estimations characterized by large uncertainties, an alternative approach is adopted. It consists of providing an estimation of the *CAPEX* of the reference case (*Concept 0*) and performing a comparative analysis for the other concepts.

In the specific case study, only the costs of the equipment to install in Johan Sverdrup are considered (6 x GE LM6000 PF and 2 x WHRU). The equipment from Edvard Grieg and Gina Krog (3 x GE LM2500+G4 and 2 x WHRU) is not considered because, at the time of the analysis, the capital investment was already been made (the offshore installations are currently operating). Since our analysis starts from 2019, the first year with expected power sent offshore, those costs cannot be avoided and are common to all the concepts. Table 2 shows the terms included in the *CAPEX* of the base case. The estimations of the costs are made in accordance with input information provided from industry as well as considering scaling effects. The *NPV* can then be calculated for *Concept 0* (i.e., $NPV_0$) according to equation above.

**Table 2.** CAPEX breakdown for *Concept 0*.

|  | Major Equipment | Installed Units | Unit Cost (M€) | Total Cost (M€) |
|---|---|---|---|---|
| *Concept 0* | LM6000 PF | 6 | 23.2 | 139.4 |
|  | WHRU | 2 | 2.0 | 4.0 |
|  |  |  | CAPEX → | 143.4 |

A different term is calculated for the other two concepts, namely the maximum capital investment ($CAPEX_{max}$). The $CAPEX_{max}$ is the capital investment which would return the *NPV* as *Concept 0*. It represents the maximum amount of capital investment for a specific concept, which would return a better economic performance compared to the reference case. If a concept can be developed with a capital investment lower than its $CAPEX_{max}$, such concept will entail a better economic performance in comparison to the base case ($NPV > NPV_0$). Vice versa if the capital investment is higher than the $CAPEX_{max}$. Whenever a reliable *CAPEX* estimation is made available for a given concept, it becomes straightforward to evaluate its economic impact by checking whether its value is lower or higher than the $CAPEX_{max}$.

The one described is a simplified economic analysis. It takes into account only the costs directly related to the power generation unit, while the many other costs associated with the operation of such complex systems are not considered as well as the revenues associated to the sale of the produced hydrocarbons. The underlying assumption is that the remaining units of the offshore plant were unaffected by the method used to supply energy. This is in line with the objective of the work, which aims to compare different concepts for providing heat and power offshore. Another limitation is that the model does not quantify the economic advantages given by difference levels of plant stability, availability and maintainability specific to the different concepts.

## 5. Results

In this section, the main results are presented and discussed. First, the results from the power system modelled are presented, followed by the results of the integrated analysis. The results of the process model are not presented in a dedicated section but included in the integrated analysis.

### 5.1. Results From the Power System Model

Simulations were performed for different years from 2015 to 2060, representing the status of the energy system. With changing prices from the three IEA scenarios (CP scenario, NP scenario, 450 scenario), the generation mix of the energy system changes accordingly, even if the power system behind stays the same. The change results in a different use of the available power plant portfolio. Expensive plants are not used, if their marginal costs are too high. For the chosen years, the generation mix of the European system and the resulting $CO_2$ emissions are shown in Figure 8.

The energy produced from solar and wind power, nuclear and biomass power plants as well as the electrical energy from CHP-plants can be seen as nearly the same as given by the sources for the input data [33,35]. This is due to the very low or even zero running costs for production (renewables have no $CO_2$-tax, base load plants have low fuel prices). As a result, the two main "free variables" in the system are the coal and gas power plants. Since EMPS aims for the maximization of the socio-economic benefit, the marginal production costs (fuel + taxes) of these two power plant types mainly determine the generation mix and the outcome of the different IEA price scenarios. This can be seen in the Figure 8, where the generation mix for the three price scenarios only differs in the coal and gas production, resulting in different $CO_2$-emissions. Zooming into one year from Figure 8, the generation mix for each country is shown in Figure 9.

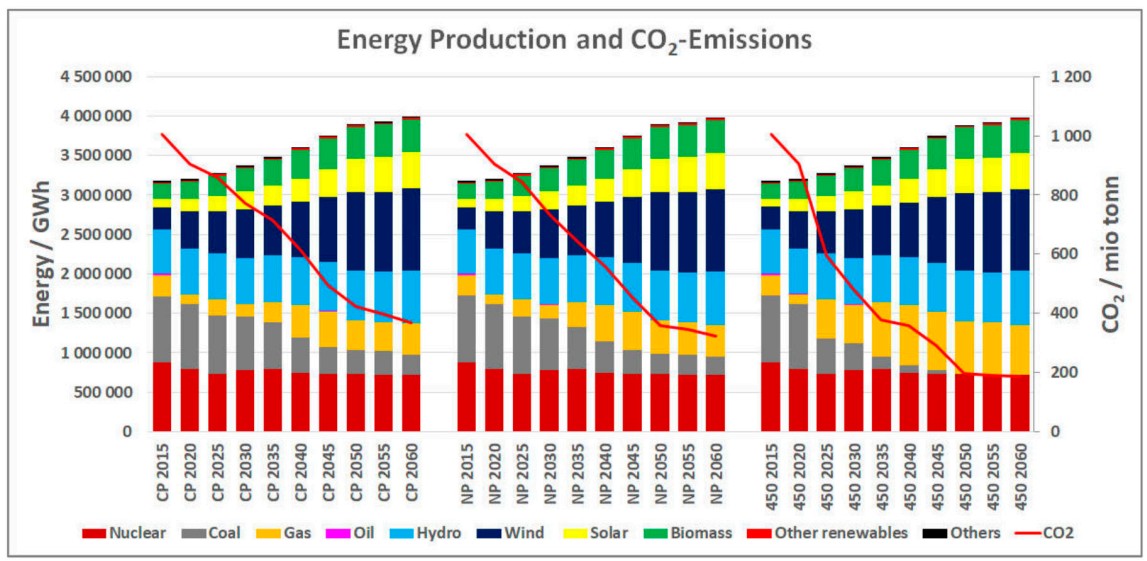

**Figure 8.** Production mix and $CO_2$ emissions in the three IEA price-scenarios for the different years.

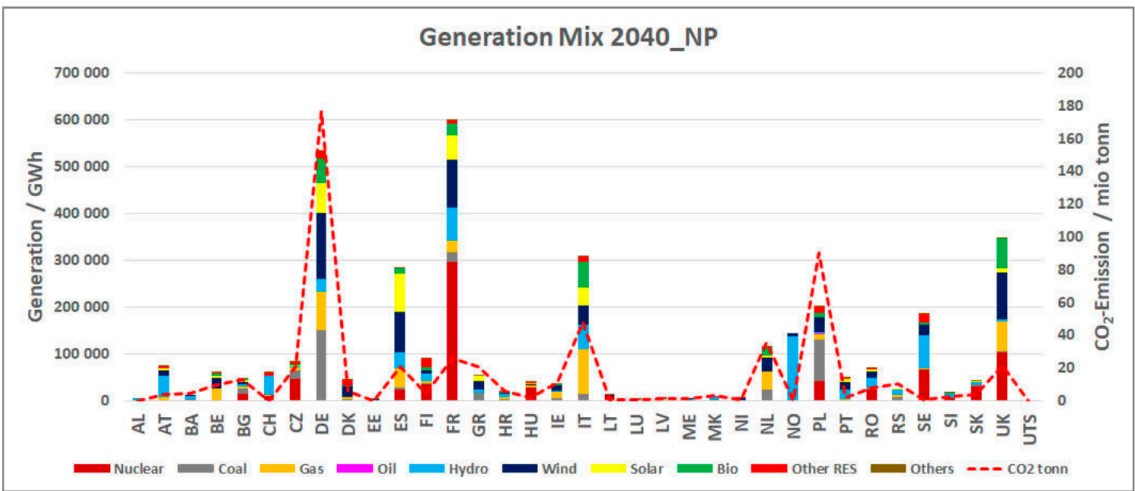

**Figure 9.** Generation mix and emissions for all countries in the NP scenario for the year 2040. The following countries and power consumers are included: Albania (AL), Austria (AT), Bosnia and Herzegovina (BA), Belgium (BE), Bulgaria (BG), Switzerland (CH), Czech Republic (CZ), Germany (DE), Denmark (DK), Estonia (EE), Spain (ES), Finland (FI), France (FR), Greece (GR), Croatia (HR), Hungary (HU), Ireland (IE), Italy (IT), Lithuania (LT), Luxembourg (LU), Latvia (LV), Montenegro (ME), Macedonia (MK), Northern Ireland (NI), The Netherlands (NL), Norway (NO), Poland (PL), Portugal (PT), Romania (RO), Serbia (RS), Sweden (SE), Slovenia (SI), Slovakia (SK), United Kingdom (UK) and Utsira High area (UTS).

As shown, the marginal costs of energy production are essential for the generation mix. With the electrification of oil platforms, more energy is needed from the system. The marginal generation, i.e., how much extra energy has to be produced in the system, is also influenced by the prices in the system. Other influences are the location (energy should be produced where it is consumed), limitations (transmission lines and power plant capacities) and price differences between the areas. In the case of this study, extra energy is consumed in Norway. This leads to less export of Norwegian hydropower to the neighboring countries (since the maximal hydropower production in Norway is already optimized). With less energy imported from Norway, the countries have to increase their own production (mainly Germany, the UK and the Netherlands) or import energy from other areas (e.g., Poland, Czech Republic). The marginal generation for all the different scenarios—when the additional power demand of Utsira

is considered—is shown in Figure 10. The generation mix differences mostly come from gas and coal power plants, since other power plants are already used optimally (hydropower, biomass) or cannot produce more (wind, solar).

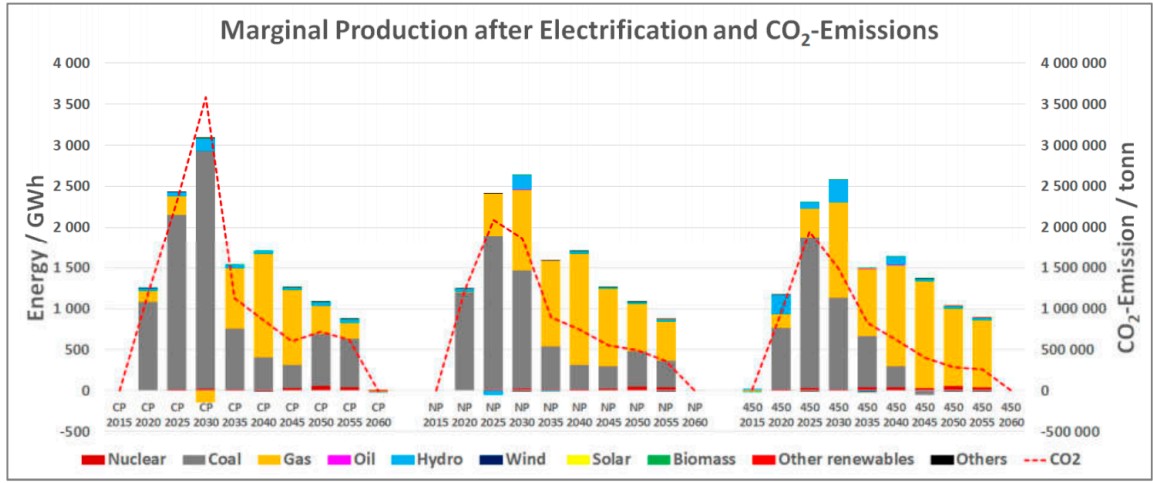

**Figure 10.** Comparison of the marginal generation of all results after the electrification.

### 5.2. Results from the Integrated Analysis

Initially, it is of interest to examine the $CO_2$-factors related to power from shore with respect to that related to the standard concept to supply energy offshore (i.e., *Concept 0*). The comparison provides a first snap-shot on the effectiveness of electrification as a mean to reduce the environmental impact of the offshore sector.

Figure 11 is informative with regard to this. It shows the $CO_2$-factors calculated with the different approaches and adjusted for offshore supply. In the figure, the diamonds represent the actual values obtained by the simulations. The line is the interpolation (or extrapolation) between the values. The results reported refer to the NP scenario. In addition, the $CO_2$-factors resulting from the utilization of offshore GTs in *Concept 0* are reported as basis for comparison (obtained from the process simulations).

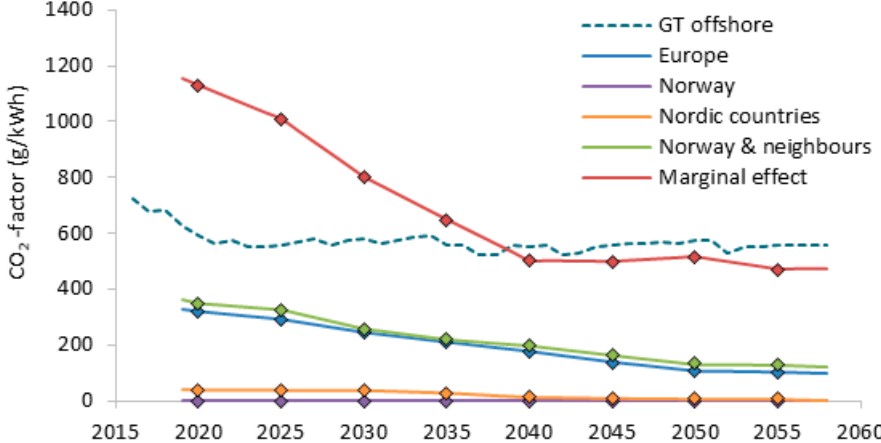

**Figure 11.** $CO_2$-factors for the NP scenario calculated with the different approaches selected and adjusted for offshore supply. The dashed line is the $CO_2$-factor related to offshore gas turbines.

If the annual $CO_2$-factor associated with PFS is higher than the corresponding value for the offshore GTs, the utilization of PFS is actually responsible for a higher rate of $CO_2$ emissions. The opposite is true when the PFS $CO_2$-factor is lower than the offshore GTs $CO_2$-factor. A significant difference can be noted between the two approaches to calculate the PFS $CO_2$-factors. With the average method, the $CO_2$-factor of PFS are always lower than those related to GTs operation, even when the near-future European



power mix is considered. The extent of the gap depends on the geographical scope considered, being particularly large for Norway or the Nordic countries, and smaller for Europe or Norway and neighbours. Independently of that, a substantial reduction of emissions related to the offshore installation can be expected by applying the average method. Conversely, the marginal method returned annual $CO_2$-factors visibly higher for the first 20 years of the analysis, because coal often becomes the marginal generator given the combination of low fuel price and low $CO_2$-price in the first years. Those years are characterized by the largest energy demands from the offshore installations and, thus, weigh more on the overall balance. After 2040, the values of the annual $CO_2$-factors are similar to those related to GTs operation, as natural gas gradually takes over as the marginal generator in the system. Over the time span, the utilization of PFS is expected to entail higher emissions when the marginal method is applied. Another element that should be pointed out is that the heat contribution to the emissions was not considered in the $CO_2$-factors associated with PFS, while that is inherently included in the $CO_2$-factors associated with GTs (since the heat is produced by exploiting the thermal energy available in the exhaust gases). The inclusion of the emissions resulting from heat production would further benefit the utilization of GTs over PFS, though the effect is expected to be relatively small.

The complete results from the environmental analysis are reported in Figures 12 and 13 that show the cumulative $CO_2$ emissions obtained with the marginal effect and with the average effect for *Concept 0* (Gas Turbine Cycles), *Concept 1* (Full Electrification) and *Concept 2* (Partial Electrification and Gas Turbine Cycles).

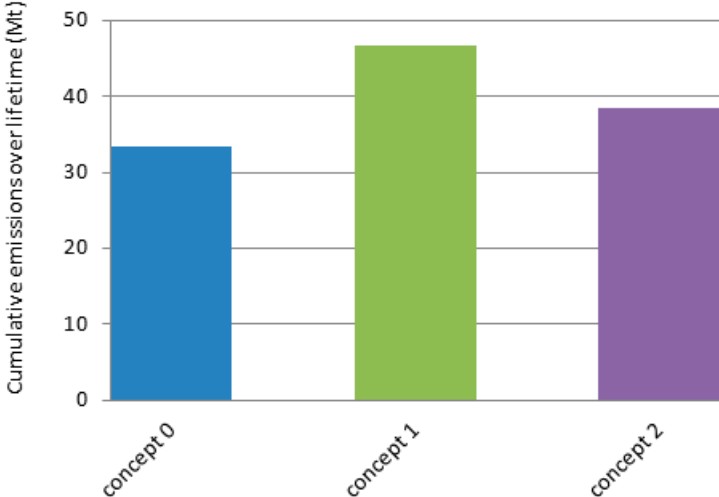

**Figure 12.** Cumulative $CO_2$ emissions over the lifetime with the marginal method from the selected concepts.

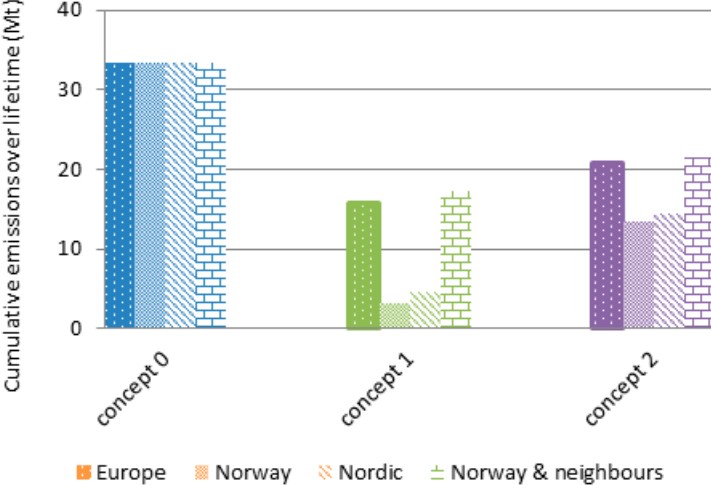

**Figure 13.** Cumulative $CO_2$ emissions over the lifetime with the average method from the selected concepts.

As it was foreseen, the marginal method penalized the concepts involving PFS. The full electrification (*Concept 1*) resulted in an increase of the cumulative $CO_2$ emissions of almost 40% (from 33.3 to 46.6 $Mt_{CO2}$). The cumulative $CO_2$ emissions of the other concept partially relying on PFS —*Concept 2*—showed an increase of 15% compared to *Concept 0*. The situation reverses when the average method is accounted for to evaluate the $CO_2$-factor. In this case, the results for different geographical scopes are reported. The result from *Concept 0* remains constant, given that it does not rely on PFS to any extent. The average method makes the utilization of PFS advantageous on an environmental point of view. A unit of power is associated to a mix of energy sources, among other renewable sources, reducing its carbon footprint and making it cleaner than a unit of power that is produced offshore through GTs. The extent of this environmental benefit depends on the geographical scope selected for the average method. If the Norwegian electricity was considered, the cut in $CO_2$ emissions would be massive (up to 90% for *Concept 1*), as the Norwegian electricity is produced to a very large degree by hydropower. The selection of other system boundaries would reduce this gap. In comparison to full electrification, *Concept 2* (Partial Electrification and Gas Turbine Cycles) has a worse environmental performance, but still quite better than the base case *Concept 0* (Gas Turbine Cycles).

The second aspect studied was the economic performance of the various concepts. Figure 14 shows the discounted annual operational costs related to the concepts as a sum of the three cash flows considered (gas, $CO_2$ and power). The base case (*Concept 0*) is that characterized by the highest operational costs, mainly due to large gas consumption and onsite $CO_2$ emissions. The reduction of the operational costs seen with the other concepts is the basis for building a margin to afford a larger initial capital investment (i.e., for developing a $CAPEX_{max}$). The larger the gap between the operational costs throughout the years, the larger the $CAPEX_{max}$.

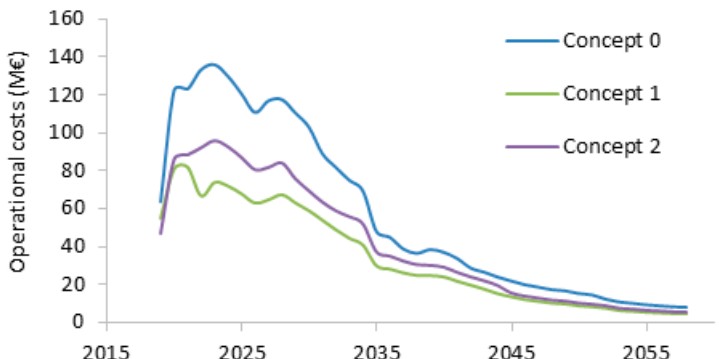

**Figure 14.** Total operational costs for the different concepts selected (NP scenario).

Table 3 reports the $CAPEX_{max}$ values obtained. The concepts studied would return a better economic performance in comparison to *Concept 0* if they could be developed with an initial additional investment no higher than their $CAPEX_{max}$. In [29] it is estimated that the electrification of the Utsira High area would require a capital investment of 12,500 ± 3750 MNOK (approximately 1340 ± 400 M€). Given a calculated $CAPEX_{max}$ of 1052 M€, the economic viability of electrification appears to be, thus, challenging. *Concept 2* displays a lower $CAPEX_{max}$, thus a relatively worse economic performance. It is difficult to envisage a potential for implementation, even though it may offer advantages taking into account elements such as sparing and flexibility. It should be stressed that the approach utilized for the economic analysis does not allow to draw conclusions regarding the economics of the various concepts as the calculation of the specific CAPEX was not attempted. However, the $CAPEX_{max}$ allows to have a first comparative evaluation on the economic performance of the concepts and can become very useful to determine the economic viability once a detailed calculation of the expected capital investment is available.

**Table 3.** $CAPEX_{max}$ for the different concepts selected (NP scenario).

| $CAPEX_{max}$ (M€) | EU ref + NP |
| --- | --- |
| *Concept 0* | - |
| *Concept 1* | 1052 |
| *Concept 2* | 759 |

## 6. Sensitivity Analysis

Given the type of analysis undertook, involving very large systems and a long time-span, a large uncertainty is expected to be associated to the results. In order to partially address this high uncertainty level, sensitivity analyses were performed and are presented in the next sections.

*Impact of Different Scenarios*

The results presented so far refer to the simulations carried out with input data and assumptions from the NP scenario, chosen as the basis for the study. However, estimating the evolution of the power generation sector in the next 40 years is a complex exercise and large uncertainties are unavoidable. Thus, two additional IEA-scenarios were considered, namely the CP scenario and the 450 scenario. While the NP scenario describes the most likely future developments in the energy sector, the CP and 450 scenarios can be seen extreme cases, characterized by a very weak or very strong commitment towards environmental issues. The spread of results obtained by simulating those limiting cases defines a range that is supposed to contain with good confidence the uncertainty related to the current analysis.

Figure 15 shows the impact of different scenarios on the obtained $CO_2$-factors adjusted for offshore energy supply (similarly to what done in Figure 11). For the sake of an explanatory representation, the outputs related to only two cases are illustrated, namely the marginal method and the average method over Norway and neighbours. The full lines represent the outputs from the NP scenario, the dashed lines from the CP scenario and the dotted lines from the 450 scenario.

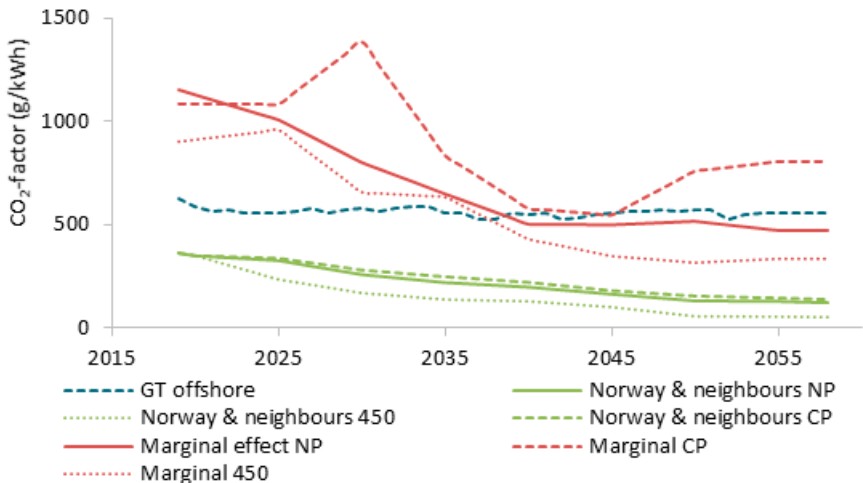

**Figure 15.** $CO_2$-factors for the future scenarios considered calculated with two different approaches and adjusted for offshore supply.

As expected, the results from the NP scenario lies in the middle, while the CP and 450 scenarios define the range within which results could be expected. The general outcome does not appear to change when different future scenarios are evaluated. The marginal method still entails larger $CO_2$-factors in comparison than offshore GT for most of the first years, under any scenario, since fossil generation remains at the marginal generator. A difference can be noted in the last years, where the 450 scenario, implying a decarbonisation of the power generation sector, resulted in significantly lower $CO_2$-factors than the offshore GTs. Conversely, the CP scenario shows $CO_2$-factors associated with the

PFS consistently higher than those from offshore GTs. When the average method is taken into account, PFS showed to be a cleaner energy source under any scenarios, with the extent of the environmental benefit depending on the scenario selected.

Figures 16 and 17 show the relative change in cumulative $CO_2$ emissions due to the implementation of *Concept 1* (Full Electrification) instead of *Concept 0* (Gas Turbine Cycles) for the three future scenarios simulated. Figure 16 considers the marginal method, while Figure 17 the average method over Norway and neighbours. The cumulative $CO_2$ emissions were estimated to increase with the marginal method from a minimum of 20% (450 scenario) to a maximum of 78% (CP scenario). The average method depicts a different situation, where the cumulative $CO_2$ emissions are substantially decreased from a minimum of 25% (CP scenario) to a maximum of 51% (450 scenario).

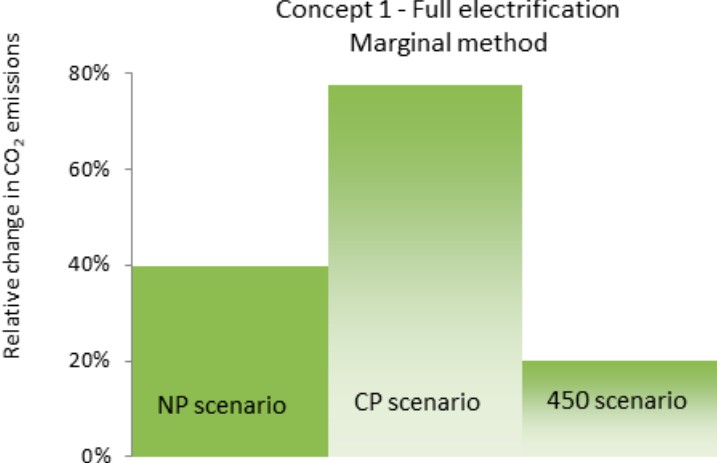

**Figure 16.** Relative change of $CO_2$ emissions of full electrification (*Concept 1*) compared to the utilization of gas turbine cycles (*Concept 0*) when the marginal method is applied to calculate the $CO_2$-factor.

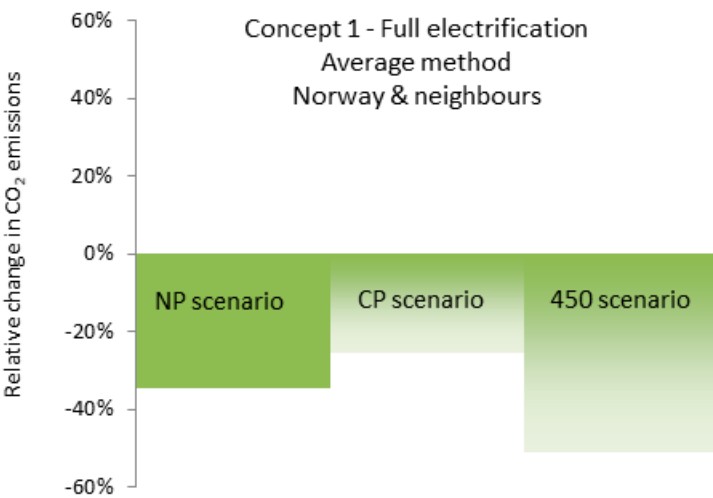

**Figure 17.** Relative change of $CO_2$ emissions of full electrification (*Concept 1*) compared to the utilization of gas turbine cycles (*Concept 0*) when the average method over Norway and neighbours is applied to calculate the $CO_2$-factor.

The effect on the economic analysis was also investigated. The outcome may seem unexpected at first sight. The scenario characterised by more stringent environmental policies (i.e., 450 scenario) was that returning the lowest $CAPEX_{max}$ (934 M€), thus it is the one where the electrification would be the most economically challenging. Even though the cost of the $CO_2$ emissions would be particularly high, such surge in $CO_2$ prices would also lead to higher costs to produce power and, consequently, to higher power prices. Offshore power generation, on the other hand, already has a high $CO_2$ tax,

which we have assumed to be rather stable throughout the years (at least in the short-run). When considering the electrification project, the higher power prices more than balance out the gains for the reduced emission costs at the offshore site. Conversely, the CP scenario obtained the highest $CAPEX_{max}$ value (1110 M€).

## 7. Conclusions

The electrification of a large offshore area in the North Sea has been thoroughly investigated and compared to alternative concepts to supply power and heat offshore. The analysis was based on an integrated model, including a process model of the offshore power generation units and a model of the European power system. The integrated assessment method allows to take into considerations a number of factors that affect the performance of an electrification project so to capture the complexity of such an analysis. The main findings of the paper are summarized below.

With respect to the environmental performance:

- The calculated impact of electrification on the total $CO_2$ emissions is strongly affected by the approach chosen to evaluate the effect of an additional power demand on the power system.
- When the marginal effect is considered, the lifetime $CO_2$ emissions associated with the operation of the offshore facilities increases with electrification up to about 40% (from 33.3 to 46.6 $Mt_{CO2}$). This is due to the large utilization of coal plants to meet the marginal increase in power demand.
- When the average effect is considered, the $CO_2$ associated with the operation of the offshore facilities decreases significantly. Emission reductions included between 48% and 90% are obtained, depending on the geographical scope selected, with cumulative $CO_2$ emissions as low as 3 $Mt_{CO2}$ being possible.

With respect to the economic performance:

- Economic viability is reached if the electrification project can be developed with an additional capital investment lower than 1052 M€. The literature reports higher economic requirements for the electrification of the offshore area (1340 ± 400 M€).

The sensitivity analysis provided the following additional information:

- Different future development scenarios in the energy sector do not change the relative outcome of the environmental analysis of electrification.
- A strong European commitment to environmental policies would make electrification more advantageous in terms of environmental impact, but also more economically challenging.

**Author Contributions:** The authors contributed as following: conceptualization, L.R., S.V., M.K. and L.O.N.; methodology, L.R., S.V., M.K. and L.O.N.; software, L.R. and S.V.; formal analysis, L.R. and S.V.; writing—original draft preparation, L.R. and S.V.; writing—review and editing, L.R., S.V., M.K. and L.O.N.; supervision, M.K. and L.O.N.; project administration, M.K. and L.O.N.; funding acquisition, M.K. and L.O.N.

**Funding:** This research was funded by Lundin Norway AS.

**Acknowledgments:** This publication has been produced with support from Lundin Norway AS within the research project "Electrification and efficient energy supply of offshore oil platforms".

**Conflicts of Interest:** The authors declare no conflict of interest. The funders had no role in the design of the study; in the collection, analyses, or interpretation of data; in the writing of the manuscript, or in the decision to publish the results.

## Abbreviations

| | |
|---|---|
| 450 | WEO "450 ppm" scenario |
| CAPEX | CAPital EXpenditure |
| CCGT | Combined Cycle Gas Turbine |
| CCS | Carbon Capture and Storage |
| CF | Cash Flow |
| CHP | Combined Heat and Power |
| $CO_2$ | Carbon dioxide |
| CP | WEO "Current Policy" scenario |
| DCF | Discounted Cash Flow |
| EMPS | EFI's Multi-area Power-market Simulator |
| ENTSO-E | European Network of Transmission System Operators for Electricity |
| ETS | Emission Trading System |
| EU | European Commission |
| GT | Gas Turbine |
| IEA | International Energy Agency |
| LHV | Lower Heating Value |
| NP | WEO "New Policy" scenario |
| NPV | Net Present Value |
| OCGT | Open Cycle Gas Turbine |
| PFS | Power From Shore |
| TYNDP | Ten-Year Network Development Plant |
| WEO | World Energy Outlook |
| WHRU | Waste Heat Recovery Unit |

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
