# Peer review of "An Integrated Assessment of the Environmental and Economic Impact of Offshore Oil Platform Electrification"

_energies, doi:10.3390/en12112114_

Round 1
Reviewer 1 Report
Authors have investigated the environmental and economic impact of offshore oil platform electrification. This might be important for north European countries.
A big question is how authors can validate the reliability of their model.
There is few grammatical mistakes, but the language is not authentic so it is needed for proofreading by native speakers
Need more detailed explanation on method. In particular, explain the difference between marginal and average methods.
Page 14 Figure 8, The lines are rather easy to compare, but is there need to make 3 individual graphs? Or just combining each other by using 3 different lines. The background data are identical.
Author Response
Reviewer 1
Authors have investigated the environmental and economic impact of offshore oil platform electrification. This might be important for north European countries.
Point 1: A big question is how authors can validate the reliability of their model.
Response 1: The reliability of the results of model is surely an important issue to address. A proper validation procedure was performed in some cases (i.e. process model) but was not possible in others (i.e. European power system). The reasons that led us to consider our model sufficiently reliable to be used in this study are outlined in the following paragraphs.
Concerning the validation of the process models, the following paragraphs can be found in the paper:
· From line 212: “The gas turbines (GT) were modelled with a data-defined model based on the curves provided by the manufacturer. Data for the entire operating range of the engines (10–100%) were retrieved for the two GTs modelled in this paper, a GE LM2500+G4 and a GE LM6000 PF. Those are aero-derivative gas turbines, typically used for offshore applications. The models were implemented in MATLAB [30] and the performance validated against the same engines in the Thermoflow library of gas turbine engines [31]. Thermoflow indicates that the maximum model errors for the two engines are lower than 0.5% for the exhaust mass flow rate, the power output and the heat rate and lower than 2.8°C for the exhaust temperature (test range for ambient temperature: -18 to 49°C). The successful validation suggests that similar uncertainty levels apply also to the model used for this study. A further validation process was carried out for the GE LM2500+G4. The reliability of the model was evaluated by verifying that the simulated performance was in good compliance with the actual performance of the same type of engine installed in Edvard Grieg”
· From line 231: “The geometry (fin sizes and spacing, and tubes sizes and spacing) was defined in order to reproduce the WHRU in Edvard Grieg and the simulated performance successfully compared to that.”
The power system model used in this study builds on a well proven and established model, called EMPS model. The EMPS model was developed by SINTEF Energy Research [32] in the 1970s and is continuously further developed since then. It is currently used by a multitude of users (e.g. transmission system operators, power producers, regulators, consulting companies and academics & research in Norway and abroad) for applications including:
· Forecasting of electricity prices and reservoir operation
· Long term operational scheduling of hydro power
· Maintenance planning (transmission or production)
· Calculation of energy balances (supply, consumption and trade)
· Utilization of transmission lines and cables
· Analyses of overflow losses, and probability for curtailment
· Analyse interplay between intermittent generation, hydropower and thermal power
· Investment analyses; system development studies
· Calculation of CO2 emissions from power generation”
The EMPS model has been as well used in several EU projects, such as SUSPLAN[1] and TWENTIES[2]. All this information argues for the reliability of the EMPS model and is included in the manuscript. The power system model was further developed for this study by increasing its scope in order to simulate the entire European energy system. Even though the updated model was not validated as such, the same framework and approaches used in the EMPS model applies. This allow building confidence in its reliability.
However, the results obtained are surely characterized by a significant uncertainty. Energy systems and offshore activities are extremely complex systems and producing long-term predictions of their evolution is challenging. To address this issue we used as inputs to the analysis the most reliable datasets available in the literature (information from the International Energy Agency, from the European Commission etc.). Moreover, we defined long-term scenarios to frame the results of our analysis. The two additional scenarios developed, other than the reference one, are thought as “extreme” cases and the obtained outcomes help showing the spread of the results that can be expected. Many times, the conclusions of the analysis remain qualitatively unvaried when those “extreme” scenarios are applied, helping further building confidence in the value of the study.
Point 2: There is few grammatical mistakes, but the language is not authentic so it is needed for proofreading by native speakers.
Response 2: The paper has been proof-read again and minor language mistakes have been corrected. Given the short period available for revising the paper there was not enough time to have it proofread by a native speaker.
Point 3: Need more detailed explanation on method. In particular, explain the difference between marginal and average methods.
Response 3: The following text has been added to the manuscript (from line 342): “In the very short-run, an increase in power demand due to offshore electrification in the Norwegian Continental Shelf will actually lead to more power production from regulated hydro power, since hydro power dominates the Norwegian generation mix. This again leads to less stored water in the reservoirs, but after a while the reservoir balance needs to be restored by increasing the power production from thermal power generators in neighbouring countries. So in the short run, an increase in the consumption in Norway leads to an increase in thermal power production. This is the marginal effect. Such approach has a potential problem in the identification of what is the “marginal demand”. Similarly to offshore electrification, other additional power demands (e.g. a new housing area, new electric cars, even a new and bigger panel oven in the living room) could be defined as “marginal demands”. The average method, assigning the same emission factor to each power demand, is a way to overcome this issue. However, it has other disadvantages. A main one is that the results are strongly influenced by the geographical scope in which the average factor is calculated. Defining a relevant boundary in a highly interconnected and dynamic system such as the European energy system is a challenging task.”
Point 4: Page 14 Figure 8, The lines are rather easy to compare, but is there need to make 3 individual graphs? Or just combining each other by using 3 different lines. The background data are identical.
Response 4: There are some small but meaningful differences in the background data between the three scenarios. The differences concerned the gas and coal electricity generation. Note for example that coal is phasing out in the 450 scenario, substituted by additional gas. Even though the trends are very similar we believe of significance to keep the three graphs to show that the results from the power system simulations have small but relevant differences when different model inputs (in this case price scenarios) are used.
[1] SUSPLAN - Development of regional and Pan-European guidelines for more efficient integration of renewable energy into future infrastructures n.d. https://cordis.europa.eu/project/rcn/90323_de.html.
[2] TWENTIES Task 16.3 "Grid restriction study: Nordic hydropower and Northern European Wind Power.

Reviewer 2 Report
Overall, I liked reading this article very much. The topic is interesting and highly relevant for the journal. The analysis, concepts and methods are properly described, and the article is well written. I think this article deserves to be accepted for publication. However, I am confused by the use and comparison between the two methods: marginal and average. As stated by the authors, the two methods diverge widely. Actually, their conclusions are not only different, they are opposite (figures 16 & 17). Lines 341-342, the authors write "it can be discussed which gives the most correct answer", and I certainly think that a discussion on this is necessary, but I failed to find it in the article. I suggest to add a paragraph on this in the discussion, because as it is now, the reader is presented with two methods that provide contradicting results, and no discussion to help assess their respective strengths and weaknesses.
Here are a few other minor comments to help improve the presentation:
- line 337: please precise what countries belong to the categories "Nordic countries" and "Norway & neighbours"
- line 366: for the first occurence of 450, maybe rewrite "the 450 (ppm) scenario"
- equation 3: the variable N is introduced but is not part of the equation
- figure 9: What does NI and UTS stand for? Maybe provide the countries' full names, either in the figure caption or in the list of abbreviations.
- figure 10: bars for CP-2030, NP-2025 & 450-2045 are not properly formatted. The order of the colours is different from the others and they start below 0 GWh
- line 487: reference problem
Author Response
Reviewer 2
Overall, I liked reading this article very much. The topic is interesting and highly relevant for the journal. The analysis, concepts and methods are properly described, and the article is well written. I think this article deserves to be accepted for publication.
Points 1: However, I am confused by the use and comparison between the two methods: marginal and average. As stated by the authors, the two methods diverge widely. Actually, their conclusions are not only different, they are opposite (figures 16 & 17). Lines 341-342, the authors write "it can be discussed which gives the most correct answer", and I certainly think that a discussion on this is necessary, but I failed to find it in the article. I suggest to add a paragraph on this in the discussion, because as it is now, the reader is presented with two methods that provide contradicting results, and no discussion to help assess their respective strengths and weaknesses.
Response 1: An additional explanation on the two methods has been added, including their pros and cons, and the reason why both were used in the study. See from line 342: “In the very short-run, an increase in power demand due to offshore electrification in the Norwegian Continental Shelf will actually lead to more power production from regulated hydro power, since hydro power dominates the Norwegian generation mix. This again leads to less stored water in the reservoirs, but after a while the reservoir balance needs to be restored by increasing the power production from thermal power generators in neighbouring countries. So in the short run, an increase in the consumption in Norway leads to an increase in thermal power production. This is the marginal effect. Such approach has a potential problem in the identification of what is the “marginal demand”. Similarly to offshore electrification, other additional power demands (e.g. a new housing area, new electric cars, even a new and bigger panel oven in the living room) could be defined as “marginal demands”. The average method, assigning the same emission factor to each power demand, is a way to overcome this issue. However, it has other disadvantages. A main one is that the results are strongly influenced by the geographical scope in which the average factor is calculated. Defining a relevant boundary in a highly interconnected and dynamic system such as the European energy system is a challenging task. The two methods apparently are very different, and it can be discussed which gives the most correct answer. In this study the outputs based on both approaches are reported. This allows to quantify the spread of results, and to gain a better understanding of the consequences of the assumptions made.”
Here are a few other minor comments to help improve the presentation:
Points 2: - line 337: please precise what countries belong to the categories "Nordic countries" and "Norway & neighbours"
Response 2: The countries have been specified (here and in the text) as following: Nordic countries (Norway, Sweden and Finland); Norway & neighbours (Norway, Sweden, Denmark, Germany, The Netherlands and United Kingdom)
Points 3:- line 366: for the first occurence of 450, maybe rewrite "the 450 (ppm) scenario"
Response 3: The suggested modification has been implemented.
Points 4:- equation 3: the variable N is introduced but is not part of the equation
Response 4: We corrected the imprecision.
Points 5:- figure 9: What does NI and UTS stand for? Maybe provide the countries' full names, either in the figure caption or in the list of abbreviations.
Response 5: The countries’ full name has been provided in a footnote.
Points 6:- figure 10: bars for CP-2030, NP-2025 & 450-2045 are not properly formatted. The order of the colours is different from the others and they start below 0 GWh
Response 6: The difference is not actually a mistake rather a different type of outcome from the marginal generation analysis. In those years, when the additional power demand for electrification is introduced, the optimization process leads to a decrease of gas power generation (CP-2030) or hydro power generation (NP-2025) or coal power generation (450-2045) compared to the reference case. Therefore the negative values.
Points 7:- line 487: reference problem
Response 7: The imprecision has been corrected.

Reviewer 3 Report
The paper is well written and the topic is of course of great interest in this period of "energy transition". The only comment suggested by the reviewer is that the references section must be improved; the following papers are suggested:
1) Leporini, M., Marchetti, B., Corvaro, F., & Polonara, F. (2019). Reconversion of offshore oil and gas platforms into renewable energy sites production: Assessment of different scenarios. Renewable Energy, 135, 1121-1132.
2) Roussanaly, S., Aasen, A., Anantharaman, R., Danielsen, B., Jakobsen, J., Heme-De-Lacotte, L., ... & Dreux, R. (2019). Offshore power generation with carbon capture and storage to decarbonise mainland electricity and offshore oil and gas installations: A techno-economic analysis. Applied energy, 233, 478-494.
3) Sedlar, D. K., Vulin, D., Krajačić, G., & Jukić, L. (2019). Offshore gas production infrastructure reutilisation for blue energy production. Renewable and Sustainable Energy Reviews, 108, 159-174.
Author Response
Reviewer 3
The paper is well written and the topic is of course of great interest in this period of "energy transition".
Point 1: The only comment suggested by the reviewer is that the references section must be improved; the following papers are suggested:
1) Leporini, M., Marchetti, B., Corvaro, F., & Polonara, F. (2019). Reconversion of offshore oil and gas platforms into renewable energy sites production: Assessment of different scenarios. Renewable Energy, 135, 1121-1132.
2) Roussanaly, S., Aasen, A., Anantharaman, R., Danielsen, B., Jakobsen, J., Heme-De-Lacotte, L., ... & Dreux, R. (2019). Offshore power generation with carbon capture and storage to decarbonise mainland electricity and offshore oil and gas installations: A techno-economic analysis. Applied energy, 233, 478-494.
3) Sedlar, D. K., Vulin, D., Krajačić, G., & Jukić, L. (2019). Offshore gas production infrastructure reutilisation for blue energy production. Renewable and Sustainable Energy Reviews, 108, 159-174.
Response 1: All the suggested references are now included in the paper.

Round 2
Reviewer 1 Report
..